# CAUSALITY-BASED BLACK-BOX BACKDOOR DETECTION

## ABSTRACT

Deep Neural Networks (DNNs) are known to be vulnerable to backdoor attacks, where attackers can inject hidden backdoors during the training stage. These attacks pose a serious threat to downstream users who unintentionally use third-party backdoored models (e.g., HuggingFace, ChatGPT). To mitigate the backdoor attacks, various backdoor detection methods have been proposed, but most of them require additional access to the model's weights or validation sets, which are not always available for third-party models. In this paper, we adopt a recently proposed setting, which aims to build a firewall at the user end to identify the backdoor samples and reject them, where only samples and prediction labels are accessible. To address this challenge, we first provide a novel causality-based perspective for analyzing the heterogeneous prediction behaviors for backdoor and clean samples. Leveraging this established causal insight, we then propose a Causality-based Black-Box Backdoor Detection algorithm, which introduces counterfactual samples as an intervention to distinguish backdoor and clean samples. Extensive experiments on three benchmark datasets validate the effectiveness and efficiency of our method. Our code is available at https://anonymous.4open.science/r/CaBBD-4326/.

## 1 INTRODUCTION

Deep neural networks (DNNs) have achieved tremendous success in various applications, such as face recognition(Kortli et al., 2020), object detection(Zou et al., 2023), and language translation(Vaswani et al., 2017; Sun et al., 2022). Despite these successes, training large DNNs needs considerable time and computational resources. Consequently, many users opt to utilize third-party pre-trained models through API requests (e.g., ChatGPT), or directly download them from online platforms (e.g., ModelZoo, HuggingFace).

However, recent research found that DNNs can be easily attacked by injecting imperceptible backdoors during the training stage(Gu et al., 2017; Chen et al., 2017; Nguyen & Tran, 2021). After backdoor injection, the DNN's prediction results can be maliciously manipulated by the adversaries whenever the input sample contains the pre-defined trigger pattern, while it behaves normally when the input sample is clean. This vulnerability can pose a serious threat to downstream users, especially for some safety-critical scenarios such as autonomous driving(Han et al., 2022; Chan et al., 2022), medical diagnosis(Feng et al., 2022) and financial fraud detection(Lunghi et al., 2023), undermining users' trust in third-party models.

To mitigate the threat, there has been a plethora of work on backdoor detection(Xiang et al., 2022; Liu et al., 2022; Zeng et al., 2021; Just et al., 2023). However, most of them require white-box access to the model weights, the model architecture, or an additional validation set, which are not always provided by third-party models. Therefore, we adopt a more practical problem setting as described in(Guo et al., 2023), dubbed as input-level black-box backdoor detection (illustrated in Figure 1): Given a well-trained third-party DNN, our objective is to function as a firewall at the user end, distinguishing whether an input image is clean or backdoored. We approve and forward the DNN's prediction results for clean images to users while rejecting those for poisoned images. The challenges of the problem stem from two aspects: ❶ *Limited information*: We are only allowed to access the inference samples and the prediction results generated by the DNNs. ❷ *Efficiency requirement*: Our detection algorithm should not significantly increase the inference time for clean

samples, as it may harm the user experience. The two challenges, along with the stealthy triggers in backdoor attacks, make the problem setting inherently challenging. Some previous methods address this problem by utilizing some intriguing properties in the backdoor samples  Guo et al. (2023). However, their methodology only works with specific types of attack (e.g., patch-based attacks).

To this end, we first consider a nontrivial question that is fundamental to our analysis: What is the inner mechanism that makes a backdoored DNN consistently predict the target label when input with backdoor samples, but behave normally when input with clean samples? A naïve answer would be that the backdoored model has learned a mapping from the trigger pattern to the target label during the pre-training stage. However, this explanation only scratches the surface, offering limited insight into the design of a backdoor detection algorithm.

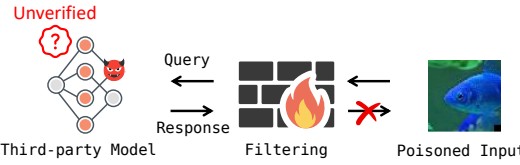

Figure 1: An illustration of the black-box input-level backdoor detection.

To step further, we introduce causal inference as a novel perspective to unravel the mechanism of heterogeneous prediction behaviors between backdoor samples and clean samples. We find that backdoor attacks act as a **confounder** (Figure 2), opening a spurious path from backdoor samples to the prediction results. Consequently, predictions on backdoor samples are predominantly led by this spurious path, while those on clean samples are led by the original causal path. Although this causal analysis provides insights into distinguishing backdoor samples and clean samples, it is challenging to explicitly identify the cause of prediction on each sample, namely which path the model follows. Hence, we propose an indirect method that implicitly induces some distinguishable behaviors of backdoor samples and clean samples due to their distinct causality. Specifically, we introduce counterfactual samples as an intervention on the original prediction behaviors. Due to the inherently different causes for predictions on backdoor samples and clean samples, the intervention on them will result in distinct observations. We could then distinguish them based on the observations. Finally, we derive a novel backdoor detection algorithm called **Ca**usality-based **B**lack-Box **B**ackdoor **D**etection (CaBBD), which employs counterfactual samples as interventions on the prediction behaviors to distinguish backdoor samples and clean samples. Specifically, we construct counterfactual samples by progressively adding noise. Extensive experiments have shown the effectiveness and efficiency of our CaBBD algorithm.

- **Novel Causality-based Perspective**. To the best of our knowledge, this is the first paper that analyzes and exploits the distinct prediction behaviors of clean and backdoored images from a causal perspective.
- **Counterfactual Backdoor Detection Algorithm**. We introduce a noise-based image perturbation method to construct counterfactuals and show its efficacy in distinguishing between backdoored images and clean images.
- **SOTA Performance in Effectiveness and Efficiency**. Various experiments across all popular datasets have empirically proven that our algorithm effectively and efficiently detects the backdoor samples in the input-level black-box setting with limited information.

## 2 PRELIMINARIES

### 2.1 MAIN PIPELINE OF BACKDOOR ATTACKS

Let $\mathcal{D} = \{\boldsymbol{x}_i, y_i\}_{i=1}^n$ denote the original dataset, where $\boldsymbol{x}_i \in \mathcal{R}^n$ denotes the image sample and $y_i$ denotes the corresponding ground-truth label. The deployed neural network model is denoted as $f_\theta$, with $\theta$ as the trainable parameters. Then the malicious backdoor attacker selects a subset of the original dataset (denoted as $\mathcal{D}_c$) and modifies it to a poisoned version with $\mathcal{D}_b = \{(\hat{\boldsymbol{x}}_i, y_t) | \hat{\boldsymbol{x}}_i = \boldsymbol{x}_i + \boldsymbol{t}_i, \boldsymbol{t}_i = \delta(\boldsymbol{x}_i), \forall (\boldsymbol{x}_i, y_i) \in \mathcal{D}_c\}$, where $y_t$ denotes the target label, $t_i$ denotes the trigger pattern for the $x_i$ and $\delta(\cdot)$ is a pre-defined trigger generation function. For example, BadNet Gu et al. (2017) adopts a constant function to generate square pixel patterns for each of the poisoned samples. Following the taxonomy in Li et al. (2022), we categorize backdoor attacks into "sample-agnostic"

and "sample-specific" based on the properties of trigger function $\delta(\cdot)$ and provide the corresponding formal definitions as follows:

**Definition 1** (Sample-agnostic Trigger). *The trigger function $\delta(\boldsymbol{x})$ of sample-agnostic backdoor attacks is a constant function, namely $\delta(\boldsymbol{x}) = c$, where $c$ is a fixed value for all input samples $\boldsymbol{x}$.*

**Definition 2** (Sample-specific Trigger). *The trigger function $\delta(\boldsymbol{x})$ of sample-specific backdoor attacks is an injective function, namely $\delta(\boldsymbol{x}_i) \neq \delta(\boldsymbol{x}_j), \forall \boldsymbol{x}_i \neq \boldsymbol{x}_j$.*

Then the neural network model $f_\theta$ is trained on the mixture of poisoned subset $\mathcal{D}_b$ and the remaining clean subset $\mathcal{D}_{/c}$ with the following optimization problem

$$\min_\theta \sum_{i=1}^{|\mathcal{D}_b|} \ell(f_\theta(\hat{\boldsymbol{x}}_i), y_t) + \sum_{i=1}^{|\mathcal{D}_{/c}|} \ell(f_\theta(\boldsymbol{x}_i), y_i), \tag{1}$$

where $\ell(\cdot)$ denotes the loss function. In the inference stage, the DNN is expected to exhibit normal behavior when the input images are benign, but consistently predict the target labels when the trigger is present.

## 2.2 PROBLEM DESCRIPTION

In this paper, our setting is formulated similarly to that in(Guo et al., 2023; Gao et al., 2021), dubbed "input-level black-box backdoor detection". Specifically, two parties are considered: Attacker and Defender. A detailed description of each party is given as follows.

**Attacker.** The attacker aims to implant backdoors to the victim model and subsequently release it on online platforms, where downstream users could directly download the models or access the model through API requests.

**Defender.** The defender aims to establish a firewall at the user end to perform *effective* and *efficient* backdoor detections. Effectiveness requires accurately identifying whether an input image is malicious or not, while efficiency requires that the filtering method does not significantly impact the response time of user queries. The defender is assumed to be only accessible to the input images and the prediction labels provided by the DNN, with no prior information regarding the backdoor attacks or the model. With the setting in hand, our problem could then be formally stated as follows,

**Problem 1** (Input-level Backdoor detection). *Given a DNN $f_\theta(\cdot)$, the defender aims to build an algorithm $\mathcal{A}(\cdot)$ such that $\mathcal{A}(\boldsymbol{x}_{backdoor}) = 1$ when the input image contains the trigger pattern and $\mathcal{A}(\boldsymbol{x}_{clean}) = 0$ when the input image is clean.*

The challenge of the problem 1 stems from two factors: ❶ **Limited information**. $\mathcal{A}(\cdot)$ only has access to the query image and the prediction labels returned by DNN. *How to utilize the limited information and launch an effective detection algorithm*? ❷ **Efficiency guarantee**. Our detection algorithm is expected not to significantly impact the inference efficiency. *How could we design an algorithm that satisfies the two requirements simultaneously*? We will answer the two questions in the remainder of the paper.

## 2.3 BACKDOOR ATTACKS: A CAUSALITY-BASED PERSPECTIVE.

Before delving into this complex issue, let's first address a foundational question that is crucial to our analysis: What is the inner mechanism that makes a backdoored DNN consistently predict the target label when input with backdoor samples but behave normally when input with clean samples? A naïve answer would be that the backdoored model has learned a mapping from the trigger pattern to the target label during the pre-training stage. However, this explanation only scratches the surface, offering limited insight into the design of a backdoor detection algorithm. Advancing beyond this, we propose to analyze distinct prediction behaviors of clean and backdoored images from a novel causality-based perspective, leveraging the capability of causal inference to uncover core mechanisms in machine learning (Xiao et al., 2023; Zhang et al., 2023). Specifically, we introduce causal graphs in Figure 2 to model the underlying mechanism of DNN's prediction

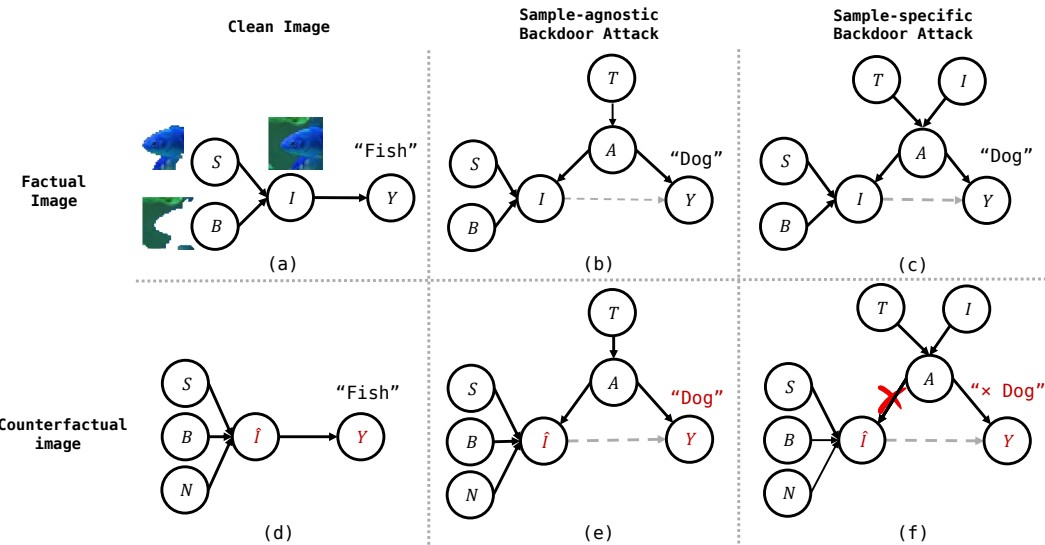

Figure 2: Causal graphs of the DNN's predictions for (a) clean images, (b) backdoor images with sample-agnostic trigger, (c) backdoor images with the sample-specific trigger, (d) counterfactual of clean images, (e) counterfactual of backdoor images with sample-specific trigger (f) counterfactual of backdoor images with the sample-agnostic trigger.

when input with different types of images, where causal graphs are directed acyclic graphs denoting causal relationships between variables. The details of the causal graphs are given as follows:

**Clean Samples.** As shown in Figure 2(a), the predicted label $Y$ of a clean image is determined by the image content $I$ ($I \to Y$), which encompasses both semantic features $S$ and background features $B$, denoted as $S \to I \leftarrow B$. For instance, consider an image of a fish, where pixels related to the fish per se are the semantic features $S$, and pixels related to the water and aquatic plants are background features $B$. A well-trained DNN will predict the image by leveraging all information contained in $I$.

**Sample-agnostic Backdoor Attacks.** As shown in Figure 2(b), valid sample-agnostic attacks $A$ depend solely on the trigger pattern $T$ while independent of image content, denoted as $T \to A$. This dependency indicates that a universal trigger can poison any clean image. These attacks $A$ modify images $I$ by injecting triggers and altering image labels to the target label $Y_t$, represented as $I \leftarrow A \to Y$. This introduces a spurious path from $I$ to $Y$, which lies outside the direct causal path ($I \to Y$). The attacks thereby serve as a **confounder**, which builds and strengthens the erroneous correlations between the modified images and the target label. Once DNNs are trained on these poisoned images, they become backdoored. Consequently, predictions for poisoned images are predominantly led by this spurious path ($I \leftarrow A \to Y$) Du et al. (2021), while the direct causal path plays a minor role, symbolized by a gray dotted line in Figure 2.

**Sample-Specific Backdoor Attacks.** Compared to sample-agnostic attacks, sample-specific attacks, shown in Figure 2(c), uniquely depend on both the trigger $T$ and the images $I$. This dependency suggests that each image possesses a unique trigger; a trigger valid for one image is not applicable to another.

The above analysis provides an intuition to distinguish backdoor samples and clean samples by assessing whether the model's prediction is primarily influenced by the direct causal path or the spurious path. However, the black-box nature of DNNs and the limited available information in our setting render this direct causal analysis challenging. Hence, we propose an indirect method that implicitly induces some distinguishable behaviors of backdoor samples and

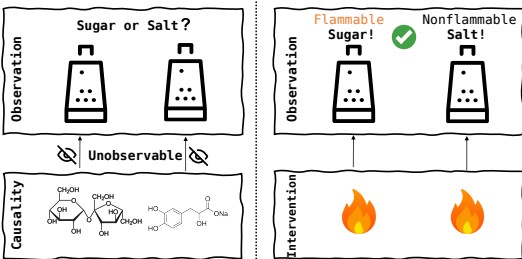

Figure 3: Example of identifying sugar and salt.

clean samples due to their distinct causality. Specifically, we introduce counterfactual samples as an **intervention** on the original prediction behaviors, where the counterfactual sample is generated by applying specialized modifications to the original factual input. Due to the inherently different causality for predictions on backdoor samples and clean samples, the intervention on them will result in distinct observations. The intuition of counterfactual intervention is motivated by a real-life example of *identifying salt and sugar* (Figure 3): Distinct molecular structures (**causality**) determine whether a substance is sugar or salt. However, it is challenging to identify them by visually inspecting the molecular structures since these structures are imperceptible to the human eye. Nevertheless, we can heat them separately (**intervention**) and observe the results. Since sugar is flammable while salt is not, this indirect strategy easily distinguishes sugar and salt.

## 3 METHODOLOGY

### 3.1 IMAGES WITH ADDITIVE NOISE ARE COUNTERFACTUAL EXAMPLES

Based on the previous analysis, the main question is now narrowed down to the following: *how to design an ideal counterfactual generation strategy that enlarges the difference between backdoor and clean samples*? Random noise has been one of the most popular methods for generating counterfactual samples. Therefore, it is natural to hypothesize that random noise might be helpful in distinguishing backdoor samples from clean samples. Besides, we also investigate other common counterfactual generation methods and provide comparative experiments. We now formalize the idea and validate the hypothesis with causal analysis.

We first define a magnitude set $\mathbb{S}$, e.g., $\mathbb{S} = \{0.6, 1.2, 1.8, ..., 6.0\}$. Then, for each element $\alpha^j$ in $\mathbb{S}$, we construct a modified sample $\tilde{\boldsymbol{x}}_i^j = \boldsymbol{x}_i' + \alpha^j \cdot \epsilon$, where $\epsilon \sim \mathcal{N}(0, \mathbb{I}_{|\boldsymbol{x}_i'|})$ denotes a random Gaussian noise, $\boldsymbol{x}_i'$ denotes the input image, $n_i = \alpha^j \cdot \epsilon$ denotes the additive noise, and $\alpha^j \in \mathbb{S}$ denotes the noise magnitude. The following analysis will answer why and how the generated counterfactual samples for clean samples and backdoor samples exhibit distinct patterns in prediction results after being fed into DNN.

**Clean Images.** In Figure 2(d), upon introducing noise, the predictions of the counterfactual images are determined by the new images $\hat{I}$, which comprise the corresponding noise $N$, the original semantic features $S$, and background features $B$. Specifically, the influence of the original semantic and background features remains dominant when $a^j$ is small, leading to predictions that remain unchanged ($f_\theta(\tilde{\boldsymbol{x}}_i^j) = y_i$). However, predictions will change after introducing a sufficient amount of noise ($a^j$ being large), we further validate this phenomenon through early experiments in Section B.

**Backdoor Images.** For a backdoor image $\hat{\boldsymbol{x}}_i$, we treat the modified image $\tilde{x}_i^j$ as a combination of a new image $x_i''$ and the original backdoor trigger $\boldsymbol{t}_i$:

$$\tilde{\boldsymbol{x}}_i^j = \hat{\boldsymbol{x}}_i + n_i = \boldsymbol{x}_i + \boldsymbol{t}_i + n_i = (\boldsymbol{x}_i + n_i) + \boldsymbol{t}_i = \boldsymbol{x}_i'' + \boldsymbol{t}_i, \tag{2}$$

After adding noise to a backdoor image with **sample-agnostic triggers** (Figure 2(e)), the original valid trigger $\boldsymbol{t}_i$ remains effective for the new image $x_i''$ due to a sample-agnostic trigger can poison any clean image. As a result, the outcomes of the new backdoor images continue to be influenced primarily by the spurious path $\hat{I} \leftarrow A \rightarrow Y$. Consequently, the image's outcome remains unchanged until the image is significantly distorted by noise (e.g., for large $a^j$).

For images with **sample-specific triggers**, as depicted in Figure 2(f), where triggers are tailored to individual images, the original backdoor triggers become ineffective for new images. As a result, the backdoor path $(A \rightarrow \hat{I})$ for new images is severed, with outcomes now primarily influenced by new images $\hat{I}$. Consequently, the outcomes of images promptly deviate from the original target label $y_t$ upon adding noise.

In summary, **images with sample-specific triggers witness immediate prediction flipping upon introducing noise, whereas clean images experience gradual outcome changes in response to noise intensity. Images with sample-agnostic triggers, however, maintain stability even with considerable added noise.** A preliminary experiment in Appendix B substantiates the phenomenon

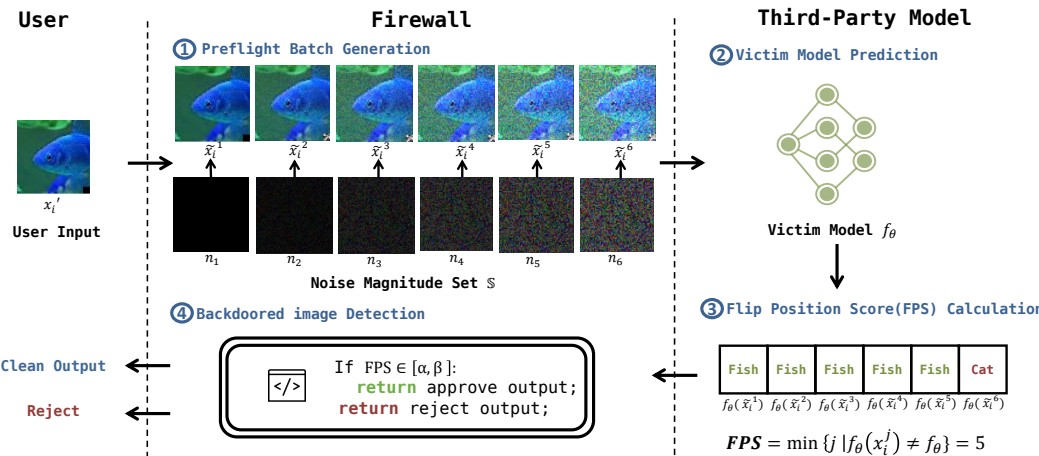

Figure 4: Pipeline of CaBBD. Upon receiving a target image $x'_i$ from the user, CaBBD first generates a counterfactual 'preflight batch' by incrementally introducing noise to $x'_i$. These counterfactual images are then fed to the DNN $f_\theta(\cdot)$ to obtain predictions. Subsequently, the FPS score is computed based on these predictions. Finally, CaBBD employs the FPS score to discern and decline queries from identified backdoor images while approving and outputting predictions for clean images.

mentioned above. Hence, we can distinguish clean samples and backdoor samples by the different positions of prediction flipping in terms of counterfactual samples.

## 3.2 THE PIPELINE OF CABBD

With the proposed counterfactual analysis, a straightforward method for detecting backdoor samples is as follows: We can progressively add random noise to the input image and use the maximum noise magnitude that flips the prediction results to determine whether the input is clean or not.

To put it more formally, given the DNN $f_\theta(\cdot)$ and a test sample $x_i$, we obtain a batch of counterfactual samples by progressively adding noise to the test sample, denoted as the "preflight batch", consisting of modified images: $P_i = \left[\tilde{x}_i^1, \tilde{x}_i^2, ..., \tilde{x}_i^{|\mathbb{S}|}\right]$. After querying the deployed DNN with the preflight batch, we record the corresponding prediction results: $R_i = \left[f_\theta(\tilde{x}_i^1), f_\theta(\tilde{x}_i^2), ..., f_\theta(\tilde{x}_i^{|\mathbb{S}|})\right]$. Then the algorithm computes a Flip Position Score (FPS) for each sample $x_i$ with $FPS(x_i) = \min\{j | f_\theta(\tilde{x}_i^j) \neq f_\theta(\tilde{x}_i^1)\}$. If the score of $x_i$ is within the threshold range $[\alpha, \beta]$, then it is determined as a clean sample; otherwise, it is a backdoor sample. An outline of the algorithm is presented in Algorithm 1. The overall pipeline is visualized in Figure 4. The design of the magnitude set and threshold range are given in the experimental section.

---

**Algorithm 1** The Backdoor Detection Method.

**Input:** Dataset $\mathcal{D}_{test} = \{(x_1, y_1), ..., (x_n, y_n)\}$; Target Model $f_\theta$; detection threshold $[\alpha, \beta]$; magnitude set $\mathbb{S}$.
**for** $i = 1$ **to** $n$ **do**
    Construct the preflight batch $P_i = \left[\tilde{x}_i^1, \tilde{x}_i^2, ..., \tilde{x}_i^{|\mathbb{S}|}\right]$ for each $(x_i, y_i) \in \mathcal{D}_{test}$ given $\mathbb{S}$.
    Obtain the prediction result for the query $P_i$, $\left[f_\theta(\tilde{x}_1^i), f_\theta(\tilde{x}_2^i), ..., f_\theta(\tilde{x}_{|\mathbb{S}|}^i)\right]$.
    Compute the score on $x_i$ following Algorithm 2.
**end for**
Filter backdoor samples by threshold $[\alpha, \beta]$.

---

**Algorithm 2** FPS Score Calculation.

**Input:** Prediction results $f_\theta(P_i) = \left[f_\theta(\tilde{x}_1^i), f_\theta(\tilde{x}_2^i), ..., f_\theta(\tilde{x}_{|\mathbb{S}|}^i)\right]$
FPS = $|\mathbb{S}|$
**for** $j = 1$ **to** $|\mathbb{S}|$ **do**
    **if** $f_\theta(\tilde{x}_j^i) \neq f_\theta(\tilde{x}_1^i)$ **then**
        **break**
    **else**
        FPS = j
    **end if**
**end for**
**return** FPS

---

Table 1: Comparison of the proposed method with other baseline defense methods in terms of precision(P), Recall(R), and AUROC(AUC).

| Dataset | Attack Method ↓ | STRIP [1] | | | Frequency [2] | | | LAVA [2] | | | Scale-UP | | | **Ours** | | |
|---|---|---|---|---|---|---|---|---|---|---|---|---|---|---|---|---|
| | | P | R | AUC | P | R | AUC | P | R | AUC | P | R | AUC | P | R | AUC |
| CIFAR-10 | BadNet | 0.85 | 0.99 | 0.98 | 0.76 | 0.64 | 0.82 | 0.70 | 0.57 | 0.78 | **0.88** | **1.00** | **0.93** | 0.85 | **1.00** | 0.91 |
| | Blend | 0.81 | 0.82 | 0.89 | 0.82 | 0.89 | 0.90 | 0.51 | 0.45 | 0.67 | 0.55 | 0.34 | 0.56 | **0.83** | **0.97** | **0.90** |
| | WaNet | 0.40 | 0.10 | 0.46 | 0.16 | 0.04 | 0.27 | 0.36 | 0.01 | 0.71 | 0.74 | 0.80 | 0.78 | **0.81** | **0.90** | **0.95** |
| | ISSBA | 0.55 | 0.49 | 0.49 | 0.55 | 0.25 | 0.56 | 0.53 | 0.01 | 0.77 | 0.79 | 0.99 | 0.92 | **0.90** | **1.00** | **0.95** |
| | Average | 0.65 | 0.60 | 0.71 | 0.57 | 0.45 | 0.63 | 0.52 | 0.25 | 0.73 | 0.74 | 0.78 | 0.80 | **0.85** | **0.96** | **0.93** |
| GTSRB | BadNet | 0.50 | 0.98 | 0.83 | 0.53 | 0.98 | **0.95** | 0.60 | 0.46 | 0.64 | 0.66 | **0.99** | 0.87 | 0.74 | 0.99 | 0.85 |
| | Blend | 0.56 | 0.99 | 0.84 | 0.53 | **0.99** | **0.97** | 0.72 | 0.68 | 0.76 | 0.56 | 0.74 | 0.59 | **0.76** | 0.99 | 0.87 |
| | WaNet | 0.53 | 0.94 | 0.75 | 0.52 | 0.92 | 0.65 | 0.63 | 0.45 | 0.67 | 0.74 | 0.87 | 0.86 | **0.98** | 0.93 | **0.91** |
| | ISSBA | 0.50 | 0.97 | 0.72 | 0.53 | **0.99** | 0.91 | 0.54 | 0.40 | 0.58 | 0.63 | 0.89 | 0.78 | **0.75** | 0.95 | **0.93** |
| | Average | 0.53 | 0.97 | 0.78 | 0.53 | **0.98** | 0.87 | 0.63 | 0.50 | 0.66 | 0.65 | 0.88 | 0.77 | **0.81** | 0.97 | **0.89** |
| ImageNet-subset | BadNet | **0.90** | 0.67 | 0.92 | 0.77 | 0.92 | 0.84 | 0.65 | 0.47 | 0.74 | 0.50 | 0.29 | 0.60 | 0.87 | **0.98** | **0.93** |
| | Blend | 0.51 | **1.00** | 0.69 | **0.78** | 0.91 | **0.87** | 0.64 | 0.55 | 0.68 | 0.64 | 0.43 | 0.70 | 0.67 | 0.77 | 0.83 |
| | WaNet | 0.51 | 0.96 | 0.55 | 0.63 | 0.52 | 0.65 | 0.59 | 0.49 | 0.72 | 0.74 | 0.81 | 0.82 | **0.88** | **0.97** | **0.93** |
| | ISSBA | 0.57 | 0.92 | 0.60 | 0.76 | 0.42 | 0.62 | 0.58 | 0.56 | 0.55 | 0.70 | 0.87 | 0.81 | **0.89** | **0.95** | **0.94** |
| | Average | 0.63 | **0.89** | 0.68 | 0.74 | 0.69 | 0.75 | 0.62 | 0.52 | 0.67 | 0.69 | 0.62 | 0.77 | **0.78** | **0.89** | **0.88** |

[1] STRIP requires additional prediction probability information.
[2] Frequency and LAVA both require an additional validation set.

# 4 EXPERIMENTS

## 4.1 EXPERIMENTAL SETTINGS

**Datasets and Models** Following(Guo et al., 2023; Gao et al., 2019; Li et al., 2021a), we choose three popular datasets for evaluating the effectiveness of our proposed method: CIFAR-10(Krizhevsky, 2009), GTSRB(Stallkamp et al., 2012), and ImageNet-subset(Deng et al., 2009). The details of the three datasets are listed in Table 3. For CIFAR-10 and GTSRB, We train with the popular ResNet(He et al., 2015). However, for the ImageNet-subset, we opted for the EfficientNet architecture(Tan & Le, 2020) as it reports a higher accuracy.

**Attack Baselines.** We choose six backdoor attacks from the well-established recent works as our baselines: 1) BadNet(Gu et al., 2017), 2) Blend Attack(Chen et al., 2017), 3) Label-Clean backdoor attacks, 4) Dynamic Attack(Nguyen & Tran, 2020), 5) WaNet(Nguyen & Tran, 2021), and 6) ISSBA(Li et al., 2021b). All the attack baselines are implemented with the open-sourced backdoor learning toolbox(Li et al., 2023). More details for each attack method can be found in Appendix D.

**Defense Baselines.** Based on our setting, it is assumed that defenders can only access the prediction results and the input images. Therefore, we compare our method with ScaleUP(Guo et al., 2023), which perfectly fits into the setting. In addition, we compare our method with Frequency(Zeng et al., 2021) and LAVA(Just et al., 2023), which require an additional validation set, and STRIP(Gao et al., 2019), which requires additional prediction probability information from the DNN model. More details about the defense baselines can be found in Appendix E.

**Implementation Details.** The full implementation details are given in Appendix G.

**Evaluation Metrics.** Following the existing works in backdoor detection(Gao et al., 2021; Guo et al., 2021; 2023), we choose the precision (P), recall (R), and the area under receiver operating curve (AUC) as the evaluation metric.

## 4.2 MAIN RESULTS

Table 6 presents the main result, where we compare our method with other defense baselines against various backdoor attacks on three datasets. For each metric, we mark the highest value with the **bold** font and the second highest value with a underline. As the table suggests, our method achieves a promising performance on all three datasets against various attack methods. Especially for sample-specific backdoor attacks (i.e., WaNet and ISSBA), our method has been shown to be significantly better than the baseline defenses. Note that in our defense baselines, the STRIP requires additional prediction probability information from the DNN model to detect backdoor samples, while our method only depends on the prediction labels. Moreover, the Frequency and LAVA leverage an

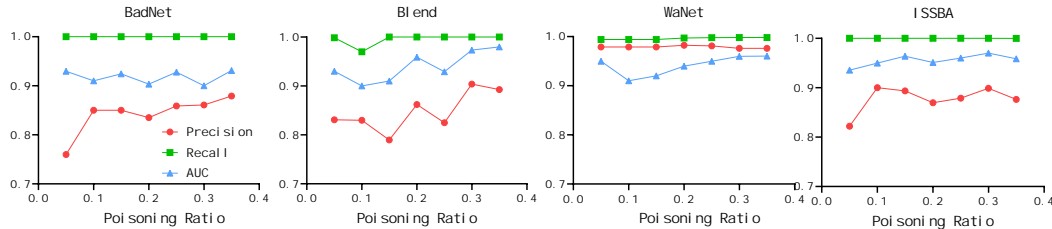

Figure 5: Performance with different poisoning ratio.

| Method | BadNet | | | Blend | | | WaNet | | | ISSBA | | |
|---|---|---|---|---|---|---|---|---|---|---|---|---|
| | P | R | AUC | P | R | AUC | P | R | AUC | P | R | AUC |
| mix-up | 0.50 | **1.00** | **0.98** | 0.52 | 1.00 | **0.99** | 0.49 | **0.98** | 0.80 | 0.50 | 0.99 | 0.76 |
| mask | 0.57 | **1.00** | 0.89 | 0.55 | **1.00** | 0.88 | 0.53 | 0.81 | 0.47 | 0.54 | 0.89 | 0.83 |
| ours | **0.85** | **1.00** | 0.91 | **0.83** | 0.97 | 0.90 | **0.81** | 0.90 | **0.95** | **0.90** | **1.00** | **0.95** |

Table 2: Comparison between other common counterfactual generation methods

additional validation set, which is also not required by our method. Our method has performed on par or even surpassed these three baselines with less information about the DNN model and the dataset. SCALE-UP is developed for the same setting as our method. However, it shows a much worse performance when defending against the Blend attack. This can be attributed to the Blend attack utilizing a global trigger (e.g., Hello-kitty-like image), while the scaling operation in the SCALE-UP can easily destroy the feature information contained in the global trigger pattern.

### 4.3 ABLATION STUDIES

**The impact of the poisoning ratio and trigger size.**    To evaluate the effectiveness of our method against different levels of poisoning ratio, we present the experimental results on CIFAR-10 in Figure 5. It is shown that our method generally achieves stable performance across different levels of poisoning ratio, suggesting that its performance is independent of the poisoning ratio. Moreover, Figure 7 shows that the performance of our detection algorithm remains stable across various trigger sizes, where the X-axis denotes the ratio of trigger size compared to the image size. The results indicate the robustness of our methods irrespective of trigger sizes.

**The impact of counterfactual generation method.**    Apart from random noise, other counterfactual generation methods, such as random masking(Xiao et al., 2023) and mixup(Yu et al., 2023), have also been widely used to generate counterfactual samples. To assess the impact of counterfactual sample design, we compare the performance of our method with mixup and random masking on the CIFAR-10 dataset and report the results in Table  2. As the table suggests, random noise shows the most stable and satisfactory performance across all types of backdoor attacks. A possible explanation is that random noise enables counterfactual sample generation with finer granularity.

### 4.4 DISCUSSION AND VISUALIZATION

**Efficiency Testing.**    Efficiency is of critical concern in our setting, since user experience is expected to not be significantly affected by the detection algorithm. Therefore, we compare the inference time before and after adopting the detection algorithms and report the results in Figure 6. The results demonstrate that our method ranks as the top-2 most efficient algorithm, exhibiting a trivial overhead compared to the vanilla inference time consumption (without detection algorithm).

**Score Distribution.**    To visually demonstrate the effectiveness of CaBBD, we plot the distribution of the FPS values for backdoor samples and clean samples in Figure 13. As the figure suggests, the FPS values for clean samples center in the middle but those for backdoor samples lie on the two sides, aligning with the causal analysis derived in the last section.

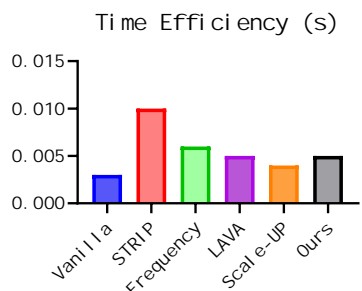
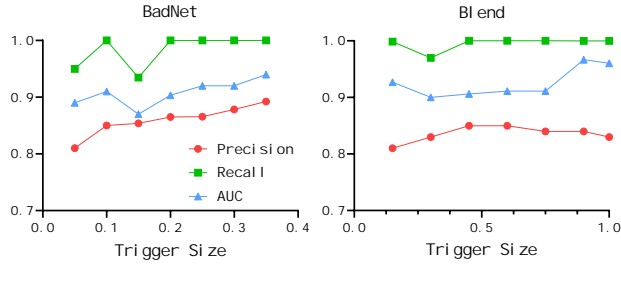

Figure 6: Comparison of inference time.    Figure 7: Performance with different trigger size.

## 5 RELATED WORK

**Backdoor Attacks**    In general, backdoor attacks can be categorized into three types: 1) data poisoning attacks(Gu et al., 2017; Chen et al., 2017; Liu et al., 2020; Guan et al., 2023b), 2) training poisoning attacks(Saha et al., 2019; Shumailov et al., 2021), and 3) model poisoning attacks(Rakin et al., 2019; Wang et al., 2022). In this paper, we solely focus on data poisoning attacks as this is the most common setting. Data-poisoning backdoor attacks aim to poison the dataset with trigger patterns. Specifically, they inject trigger patterns into the victim samples and re-assign the ground-truth label to a target label predefined by the attackers. Recent research can be divided into two categories on making the backdoor attacks more stealthy to enhance their practicality. The first one(Chen et al., 2017; Liu et al., 2020; Qi et al., 2023) aims to make the trigger pattern less visible to human eyes. For example,Chen et al. (2017) blends the clean images with random pixels. Liu et al. (2020) uses the natural reflection to construct the backdoor trigger. The other direction(Shafahi et al., 2018; Souri et al., 2021; Zeng et al., 2022) aims to make the training process less noticeable. E.g., Shafahi et al. (2018) proposes a clean-label attack, which poisons the clean images without changing labels.

**Backdoor Defenses**    Various defense methods have been proposed to mitigate the threat from the backdoor. As in(Li et al., 2022), we categorize existing defense methods into five categories. First, detection-based defenses(Gao et al., 2019; Huang et al., 2019; Guo et al., 2021; Xiang et al., 2022; Guan et al., 2023a) aim to detect whether the backdoor exists in the model. Second, preprocessing-based defenses(Doan et al., 2020) introduce a preprocessing module before the training procedure so that triggers can be inactivated. Third, defenses based on model reconstruction(Liu et al., 2018; Zhao et al., 2020) directly eliminate the effect of backdoors by adjusting the model weights or network structures. In this way, even if the trigger pattern appears, the reconstructed model will still perform normally as the backdoor is already moved. Fourth, defenses based on trigger synthesis(Wang et al., 2019; Chen et al., 2022) first reverse engineer the trigger patterns and then suppress the trigger's effects. Lastly, training sample filtering-based defenses (Li et al., 2021a; Huang et al., 2022) work by first filtering poisoned samples from the training dataset, then training the network exclusively in the rest of the dataset.

**Causal Inference and Backdoor Attacks**    To the best of our knowledge, our paper is the first to provide causality analysis for backdoor attacks in the inference stage. Despite that prior works Zhang et al. (2023) have also investigated using causal graphs in modeling backdoor attacks, our analysis fundamentally distinguishes them. A more detailed comparison of the two papers is presented in Appendix A.

## 6 CONCLUSION

In this paper, we propose an effective method for solving the input-level black-box backdoor detection problem. Our method is firstly motivated by a novel perspective for analyzing the heterogeneous prediction behaviors for backdoor samples and clean samples. Then by leveraging the causal insight, our detection algorithm introduces counterfactual samples as an intervention in the prediction behaviors to distinguish backdoor samples and clean samples. Extensive experiments across popular datasets demonstrate the effectiveness and efficiency of our method.

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

## A    COMPARISON WITH ZHANG ET AL. (2023)

Our causality analysis are fundamentally different with that in Zhang et al. (2023) with the two aspects:

**Analysis Objective**    : Their analysis aims to provide a theoretical analysis for training a clean model from a backdoored dataset, while our analysis aims to investigate the distinct prediction behaviors of clean and backdoored images from a causal perspective.

**Analysis Content**    Their analysis uses causal graphs to model the generation process of backdoor data in the **training stage**, while our analysis focuses DNNs' prediction behabiros in the **inferece stage** when input with different types of data. Although similar structures are used (e.g., $I \leftarrow A \rightarrow Y$), the actual meaning of each edge are fundamentally different. For example, in [1], $A \rightarrow Y$ means backdoor attackers will "change the labels to the targeted label" when constructing backdoor samples (evidenced by Section 3.2 of Zhang et al. (2023)), but in our setting, $A \rightarrow Y$ means that backdoor attacks will make the backdoored DNNs predict the input image as label $Y$.

**Analysis Usage**    Their analysis aims to adjust confounder, by disentangling backdoor path and causal path in the model training process. However, our causal analysis works as a guideline for distinguishing backdoor samples and clean samples in the inference stage.

## B    PRELIMINARY EXPERIMENT ON COUNTERFACTUAL EXAMPLES

**Setting**    We apply two types of backdoor attacks, including BadNet Gu et al. (2017) and WaNet Nguyen & Tran (2021) to the target model $f_\theta$. All the experiments are conducted on the CIFAR-10 dataset with ResNet18 as the neural network architecture. To fully inject backdoors into the target model, we train the neural network for 200 epochs so that the clean accuracy $\geq 90\%$ and attack success rate $\geq 98\%$. For each poisoned test set and benign test set, we record a perturbed classification accuracy, which we name as "coherence"[1], with different magnitudes of noise under each attack and plot the changing curve in Figure 8.

**Observation**    Note that the x-axis denotes the noise magnitude and the y-axis denotes the accuracy performance. Experiments under different backdoor attacks are plotted with different markers. For example, the accuracy performance on the benign test set and poisoned test set under the model poisoned with BadNet are drawn in blue and red color, respectively. As shown in Figure 8, with the increase of noise magnitude, accuracy curves for the benign test set steadily decrease, while those for the poisoned test set belong to two extremes: 1) for sample-agnostic backdoor attacks like BadNet, the accuracy performance is unlikely largely influenced by the noise magnitude, and 2) for sample-specific backdoor attacks like WaNet, the accuracy performance exhibits an abrupt reduction with only a small amount of noise. The observation just aligns with the intuition introduced in the previous section.

## C    DATASET

The details of the dataset are given in Table 3.

## D    MORE DETAILS ABOUT ATTACK BASELINES

All the attack baselines are implemented with the open-sourced backdoor learning toolbox Li et al. (2023). The details of the attack baselines are as below:

---

[1]In our current setting, where access to ground-truth labels is unavailable, we utilize the predicted labels from the test set without noise as our substitute for ground-truth labels.

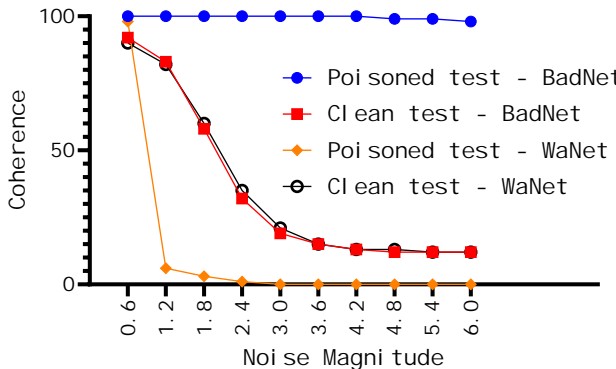

Figure 8: Coherence of the backdoored DNNs on benign testset and poisoned testset under different noise magnitudes.

Table 3: Statistical information about the Datasets

| Dataset | Image Size | # of Training samples | # of Testing Samples | # of Classes |
|---------|-----------|----------------------|---------------------|--------------|
| CIFAR-10 | $32 \times 32 \times 3$ | 50,000 | 10,000 | 10 |
| GTSRB | $32 \times 32 \times 3$ | 39,209 | 12,630 | 43 |
| ImageNet-Subset | $224 \times 224 \times 3$ | 100,000 | 20,000 | 12 |

- **BadNet** Gu et al. (2017) employs grid-like pixels as the triggers for each of the poisoned samples.
- **Blend** Chen et al. (2017) employs a hello-kitty-like image and blends it with each of the poisoned samples.
- **WaNet** Nguyen & Tran (2021) employs the interpolation method and generates sample-specific triggers for each of the poisoned samples.
- **ISSBA** Li et al. (2021b) generates sample-specific trigger patterns through an encoder-decoder network.

We present a visualization of the poisoned image generated by different backdoor attacks in Figure 9.

## E   MORE DETAILS ABOUT DEFENSE BASELINES

In this section, we introduce the basic parameter setting for each of the defense baselines.

- **STRIP** Gao et al. (2019): We follow the official implementation of STRIP[2]. Specifically, 100 samples are iteratively superimposed on the given sample and we record the classification probabilities generated by the DNN model. Subsequently, an entropy value is calculated based on the 100 probability values to determine whether the given sample is a backdoor or not. A higher entropy value denotes a higher probability of being backdoored.

---

[2]https://github.com/garrisongys/STRIP

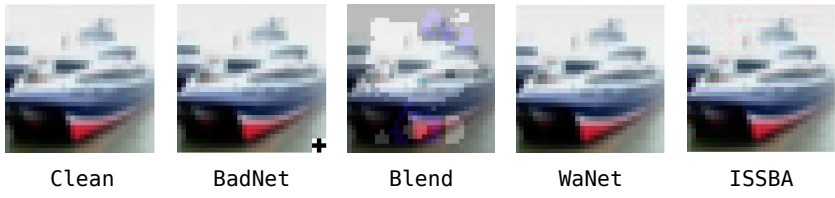

Figure 9: Visualization of backdoor samples on the CIFAR-10 dataset.

Table 4: Details about the deployed DNN models

| Dataset ↓ | BadNet | | Blend | | WaNet | | ISSBA | |
|---|---|---|---|---|---|---|---|---|
| | CA (%) | ASR (%) | CA (%) | ASR (%) | CA (%) | ASR (%) | CA (%) | ASR (%) |
| CIFAR-10 | 92 | 100 | 92 | 100 | 92 | 200 | 91 | 100 |
| GTSRB | 97 | 100 | 97 | 100 | 96 | 100 | 96 | 100 |
| ImageNet | 83 | 100 | 83 | 100 | 82 | 100 | 83 | 100 |

- **Frequency** Zeng et al. (2021): We follow the official implementation of Frequency[3]. Specifically, we employ a 6-layer CNN model as the backbone architecture of the binary detector and train it for 50 epochs on an additional validation set of 1000 samples. The binary detector determines whether the given sample is clean or backdoored by analyzing the Fourier transform of the original image. Subsequently, we use the probability of being identified as "backdoored" as the score for each sample.

- **LAVA** Just et al. (2023): We follow the official implementation of LAVA[4]. Specifically, we determine the data valuation of each data sample in the test set by calculating the proposed calibrated gradient in the original paper. A higher gradient value denotes a higher probability of being backdoored.

- **SCALE-UP** Guo et al. (2023): We follow the official implementation of SCALE-UP [5]. Specifically, the scaling set is chosen as $S = \{1, 3, 5, 7, 9\}$ for all the experiments. The proposed SPC value is calculated for each of the samples, where a higher SPC value denotes a higher probability of being a backdoor sample.

## F   MORE DETAILS ABOUT THE DEPLOYED DNN MODELS

We provide the details about the deployed DNN models in Table 4, where CA denotes the clean accuracy and ASR denotes the attack success rate. Note that for each experiment, we run the experiment three times and record the average performance of CA and ASR.

## G   MORE DETAILS ABOUT IMPLEMENTATION

Following the prior works in backdoor defences Li et al. (2021a), the poisoning ratio for backdoor attacks is set as 10% as default. The $\alpha$ and $\beta$ values are set as 1 and 6, respectively. It is noted that the design of the magnitude set $\mathbb{S}$ is a non-trivial question. If the set is too short, the granularity might not be fine-grained enough to distinguish between backdoor samples and clean samples, which is detrimental to meeting the effectiveness requirement. However, if the set is too large, the efficiency requirement cannot be satisfied. To achieve a balanced trade-off between the two sides, we have chosen 7 as a moderate length for the magnitude set, where the noise magnitude increases linearly starting from 0 with a step length of 0.2. For $\alpha$ and $\beta$, we choose 1 and 6 for all datasets, respectively. As shown in Figure 13, they are stable to distinguish backdoor and clean samples across all datasets. We also conduct sensitivity testing on different choices of step length and magnitude length $|S|$ in Figure 10-Figure 12. As seen, the choice of step length 0.2 and magnitude set 7 can generally provide a satisfactory performance against various backdoor attacks across different datasets.

## H   COMPARISON OF DIFFERENT COUNTERFACTUAL GENERATION METHODS

**The impact of counterfactual generation method**   Apart from random noise, other methods, such as random masking Xiao et al. (2023) and mixup Yu et al. (2023), have also been extensively used for generating counterfactual samples. To assess the impact of counterfactual sample design,

---

[3]https://github.com/YiZeng623/frequency-backdoor
[4]https://github.com/ruoxi-jia-group/LAVA
[5]https://github.com/JunfengGo/SCALE-UP

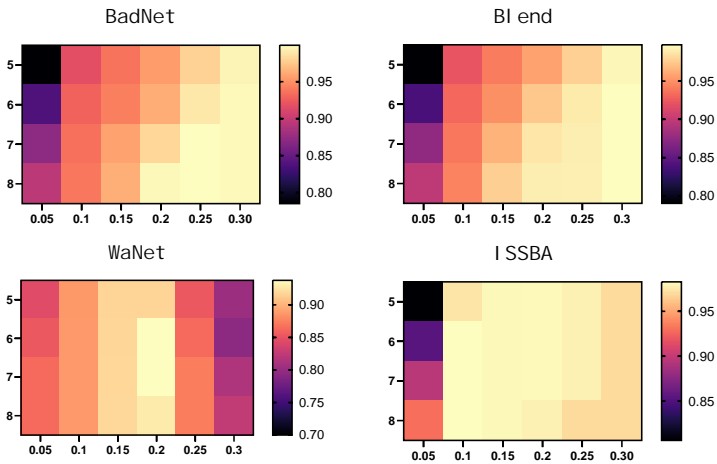

Figure 10: Sensitivity Testing on the performance of CaBBD with different choices of step length and magnitude length on Cifar10.

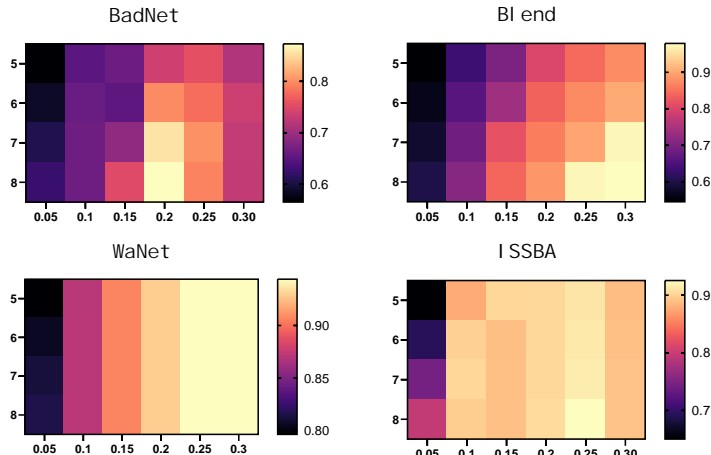

Figure 11: Sensitivity Testing on the performance of CaBBD with different choices of step length and magnitude length on GTSRB.

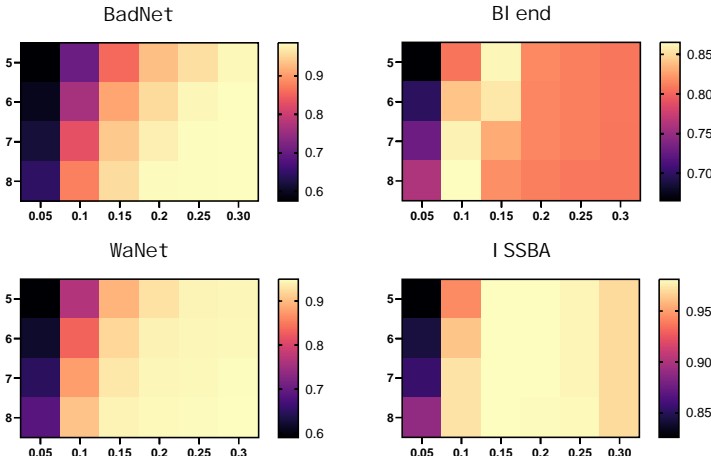

Figure 12: Sensitivity Testing on the performance of CaBBD with different choices of step length and magnitude length on ImageNet-Subset.

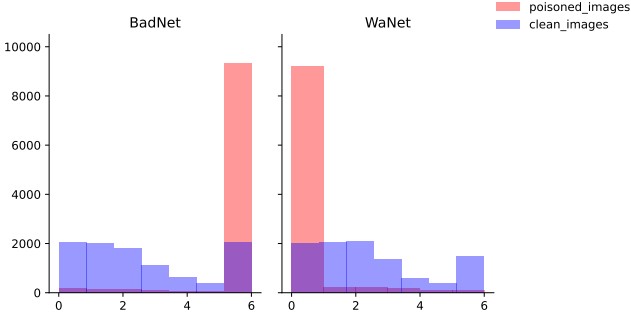

Figure 13: Comparison of FPS score distribution between poisoned and clean images

we compare the performance of our method with mixup and random masking on CIFAR-10 dataset and report the results in Table 2. As the table suggests, random noise shows the most stable performance across all types of backdoor attacks. A possible explanation is that random noise enables counterfactual samples generation with finer granularity.

## I  SCORE DISTRIBUTION

To visually demonstrate the effectiveness of CaBBD, we plot the distribution of the FPS values for backdoor samples and clean samples in Figure 13. As the figure suggests, the FPS values for clean samples center in the middle but those for backdoor samples lie on the two sides, aligning with the causal analysis derived in the last section.

## J  DISCUSSION ABOUT ADAPTIVE BACKDOOR ATTACKS

Suppose an attacker already knows our defense methods in advance, then intuitively the attacker will train the DNN model with the following adaptive training loss function:

$$\min_{\theta} \sum_{i=1}^{|D_{/c}|} \ell(f_\theta(x_i), y_i) + \sum_{i=1}^{|D_b|} \ell(f_\theta(\hat{x}_i), y_t) + \sum_{i=1}^{|D_b|} \ell(f_\theta(\hat{x}_i + m * \epsilon_i), y_i)$$

| Noise Magnitude Multiplier | 0 | | | 0.1 | | | 0.5 | | | 1.0 | | | 1.5 | | |
|---|---|---|---|---|---|---|---|---|---|---|---|---|---|---|---|
| | P | R | AUC | P | R | AUC | P | R | AUC | P | R | AUC | P | R | AUC |
| | 0.85 | 1.00 | 0.91 | 0.77 | 0.91 | 0.85 | 0.72 | 0.94 | 0.87 | 0.30 | 0.42 | 0.64 | 0.24 | 0.34 | 0.55 |

Table 5: Performance of CaBBD with different noise magnitude multiplier $m$.

Table 6: Comparison of the proposed method against more baseline attack methods.

| Dataset | Attack Method ↓ | Defense Method | | | | | | | | | | | | | | |
|---|---|---|---|---|---|---|---|---|---|---|---|---|---|---|---|
| | | STRIP | | | LAVA | | | Frequency | | | Scale-UP | | | **Ours** | | |
| | | P | R | AUC | P | R | AUC | P | R | AUC | P | R | AUC | P | R | AUC |
| CIFAR-10 | DFST | 0.51 | 0.99 | 0.70 | 0.67 | 0.54 | 0.72 | 0.5 | 1.0 | 0.45 | 0.46 | 0.55 | 0.57 | 0.78 | 0.99 | 0.90 |
| | Filter | 0.48 | 0.35 | 0.54 | 0.44 | 0.68 | 0.70 | 0.5 | 0.99 | 0.5 | 0.49 | 0.65 | 0.51 | 0.85 | 0.80 | 0.97 |
| | TaCT | 0.47 | 0.96 | 0.92 | 0.50 | 0.98 | 0.50 | 0.40 | 0.42 | 0.60 | 0.61 | 0.70 | 0.62 | 0.85 | 0.80 | 0.97 |
| | Adaptive Blend | 0.50 | 1.00 | 0.94 | 0.50 | 1.00 | 0.50 | 0.45 | 0.46 | 0.64 | 0.58 | 0.74 | 0.62 | 0.81 | 0.73 | 0.96 |
| | Refool | 0.45 | 0.32 | 0.52 | 0.50 | 1.00 | 0.81 | 0.62 | 0.69 | 0.36 | 0.80 | 0.87 | 0.85 | 0.83 | 0.88 | 0.88 |
| | SIG | 0.52 | 0.99 | 0.84 | 0.43 | 0.50 | 0.58 | 0.56, | 0.60 | 0.85 | 0.32 | 0.45 | 0.48 | 0.80 | 1.00 | 0.90 |
| | Label-clean | 0.89 | 0.45 | 0.91 | 0.5 | 0.95 | 0.85 | 0.67 | 0.61 | 0.74 | 0.76 | 0.96 | 0.91 | 0.82 | 0.97 | 0.96 |

where $\epsilon_i \sim N(0,1)$ is a random Gaussian noise added to each training sample $x_i$, and $m$ is the mangitude multiplier of the added noise. To evaluate whether our method is still effective against adaptive attacks, we provide results with varying $m$ against BadNet attack on the Cifar10 dataset in the following table.

It can be inferred that our method can still achieve satisfactory performance when the noise magnitude is small (e.g. $< 0.5$), but the performance will gradually drop when the noise magnitude multiplier becomes higher (e.g., $> 1.0$). However, we argue that when the noise magnitude multiplier ¿ 1.0, the whole image will be dominated by random noise, rendering the modified image easily detected and filtered out by the off-the-shelf backdoor filtering algorithms.

## K    MORE BASELINE ATTACKS METHODS FOR EVALUATION.

Apart from the four baseline attacks methods considered in the main experiment, we also evaluate the performance of CaBBD against other popular methods, including DFST Cheng et al. (2021), Filter Liu et al. (2019), TaCT Tang et al. (2021), Adaptive Blend Qi et al. (2022), Refool Liu et al. (2020), SIG Barni et al. (2019), and Label-clean Turner et al. (2019). The table 6 presents the results. As shown, our method has a significant advantage over the baseline defense methods in all three metrics.

