# OpenReview forum: "Causality-Based Black-Box Backdoor Detection"
_ICLR.cc/2024/Conference — Submitted to ICLR 2024_

### Official Review · Reviewer_hn12 · 2023-10-23

**Soundness:** 3 good
**Presentation:** 2 fair
**Contribution:** 2 fair
**Rating:** 6
**Confidence:** 3

**Summary:**

This paper analyzes the heterogenous prediction behaviors for backdoor samples and clean samples from the causality perspective and proposes a causality-based backdoor detection method that only requires the prediction labels from the victim model. Extensive experiments on three benchmark datasets demonstrate the effectiveness and efficiency of the proposed detection method.

**Strengths:**

- Trendy topic
- Interesting and easy-to-understand attack pipeline
- Well-written

**Weaknesses:**

- Some presentations are misleading
- More explanations are needed
- More experiments are needed

**Questions:**

- The authors leverage causal inference to find that the backdoor attacks act as a confounder, creating a spurious path from backdoor samples to the modified prediction results. Based on this insight, the authors propose a causality-based black-box backdoor detection method that employs counterfactual samples as interventions on the prediction behaviors to effectively distinguish backdoor samples and clean samples. Extensive experiments on three benchmark datasets demonstrate the effectiveness and efficiency of the proposed detection method.

- I appreciate that the paper is well-written, especially the section where the authors use causal inference to explain the different behaviors between backdoor samples and clean samples. Their detection method is interesting and easy to understand.

- In Figure 2(f), the authors aim to show that images with sample-specific triggers promptly deviate from the original prediction results by adding noise. I do not think the label here is still "Fish."

- In Section 3.1, the authors directly introduce a magnitude set by adding noise. Based on my understanding, it is necessary to explain how to choose this magnitude set, because the proposed method involves introducing varying magnitudes of noise and observing whether the prediction results are flipped in each query to conclude whether a given sample is a backdoor sample. Another alternative approach is to show that the choice of magnitude set does not affect the effectiveness of the proposed detection method.

- It would be better to conduct more experiments on the choice of $\alpha$, $\beta$, and $|S|$, and determine whether the values differ across different datasets and different attack methods.

- In Table 1, it would be better to have a notation indicating which attacks are sample-specific and which attacks are sample-agnostic. Additionally, it is unclear whether the proposed detection method performs similarly well on both types of attacks.

---

> ### Author Response · Authors · 2023-11-18
> **Response to Reviewer hn12**
>
> **Q1. More explanations on the choice of magnitude set.**
>
> Thanks for the valuable suggestion. We will answer the question with the following aspects:
>
> - [**Already Discussed: The tradeoff behind magnitude set**] Firstly, we have already briefly discussed the choice of magnitude set in the Appendix. To sum up, the choice of magnitude set is a tradeoff between efficiency and effectiveness. If the magnitude set is too short, then the granularity might not be fine-grained enough to distinguish between backdoor samples and clean samples, which is detrimental to meeting the effectiveness requirement. However, if the set is too large, the efficiency
> requirement cannot be satisfied.
>
> - [**Yet Discussed: How the parameters impact the choice of magnitude set**] To further clarify the confusion, we conduct sensitivity testing on the choice of magnitude set with the following experiment on the Cifar10, GTSRB, and Imagenet-subset. We formalize a magnitude set with two parameters: step_length and magnitude_length, where step_length is the gap between two noise magnitudes and magnitude_length is the number of noise magnitudes.
>
> We conduct sensitivity testing on the choice of magnitude set against four backdoor attack methods over three datasets. A quick view of the result is: https://ibb.co/m0m97Lm, where the x-axis (horizontal) denotes the step_length, the y-axis (vertical) denotes the length of the magnitude set, and the value in each cell denotes the AUC score. It is seen that the method performs well when we choose the magnitude set with step_length=0.2 and length=7. These results have been added to the Appendix G of the updated manuscript.
>
> **Q2. More experiments on $\alpha, \beta$.**
>
> Thanks for the insightful comment. In our setting, $\alpha$ and $\beta$ are not hyper-parameters that need to be tuned but some fixed parameters that can be pre-defined. Specifically, $\alpha$ value is fixed as 1 and $\beta$ value is fixed as $|S|-1$. We will answer the reasoning behind the choice with the following two aspects:
>
> Firstly, the choice is highly driven by our causality analysis in Section 2. Specifically, the sample-specific backdoor samples will show an abrupt prediction flipping if a small amount of noise is added, while sample-agnostic backdoor samples will show a consistent prediction until a considerably large noise is added. Therefore, $\alpha=1$ value is set to capture the sample-specific backdoor samples, and $\beta=|S|-1$ value is set to capture the sample-agnostic backdoor samples. We argue that setting $\alpha$ and $\beta$ to other values may lead to misalignment with the causality analysis.
>
> Secondly, the provided experiments across Section 4 all demonstrate that this choice is able to perform well on all three datasets against four types of backdoor attacks.
>
> To sum up, we can readily use fixed value $\alpha = 1$ and $\beta= |S|-1$ when applying our backdoor detection method in practice.
>
> **Q3. More explanations on Table 1.**
> Thanks for the contributive suggestions. In Table 1, BadNet and Blend correspond to the sample-agnostic backdoor attacks since their trigger patterns are universal, while WaNet and ISSBA correspond to the sample-specific backdoor attacks since their trigger patterns are customized for each individual sample.
>
> Therefore, the experimental results in Table 1 already validate the effectiveness of our method on both types of backdoor attacks.
>
> **Q4. Clairifcations on Figure 2(f).**
>
> Thanks for the comments. We totally agree with your point that the prediction might not be "fish". We have accordingly revised the figure in the updated manuscript.

---

> ### Author Response · Authors · 2023-11-20
> **Response to Reviewer hn12**
>
> Dear Reviewer,
>
> Thanks for your assessment and your encouragement of our work. As the deadline for reviewer-author discussion (11/22/23) approaches, we would be happy to take this opportunity to make more discussions about any of our questions or concerns. If our response has addressed your concerns, we would be grateful if you could re-evaluate our paper based on our feedback and new supplementary results.
>
> Sincerely,
>
> Authors

---

> ### Author Response · Authors · 2023-11-21
> **Looking forward to your reply!**
>
> Dear Reviewer,
>
> Thanks for your assessment and your encouragement of our work. As the **deadline for reviewer-author discussion (11/22/23) approaches**, we would be happy to take this opportunity to make more discussions about any of our questions or concerns. If our response has addressed your concerns, we would be grateful if you could re-evaluate our paper based on our feedback and new supplementary results.
>
> Sincerely,
>
> Authors

---

### Official Review · Reviewer_u6kT · 2023-10-28

**Soundness:** 3 good
**Presentation:** 3 good
**Contribution:** 3 good
**Rating:** 6
**Confidence:** 4

**Summary:**

The paper explores a black-box backdoor detection problem that can only access testing samples and labels. By analyzing the heterogeneous prediction behaviors for backdoored and clean samples, the paper proposes a Causality-based Black-Box Backdoor Detection (CaBBD) method to distinguish whether a sample is clean or backdoored. Specifically, CaBBD introduces counterfactual samples as intervention to check the difference of model outputs. Extensive experiments on three datasets and four datasets indicate the effectiveness of CaBBD.

**Strengths:**

The paper explains the intuition of proposed method from the causality-based perspective, which makes the proposed reasonable. Also, some preliminary experiments (e.g. figure 8) demonstrate the observations (at the bottom of page 5) that clean and backdoored samples can behave differently when attatching noises with different sthengths. The extensive experiments demonstrate the effectiveness of proposed method in Table 1.

**Weaknesses:**

The experiments are not sufficient. The paper presents results using four attacks including BadNet, Blended, WaNet and ISSBA in Table 1. There is no clean-label attacks such as label-clean [1], SIG [2] and ReFool [3]. It is better to show the results on clean-label attacks.

Some typos are obvious. For example, in the caption of figure 2, (b) should be sample-agnostic trigger and (c) should be sample-specific trigger.

[1] Turner, Alexander, Dimitris Tsipras, and Aleksander Madry. "Label-consistent backdoor attacks." arXiv preprint arXiv:1912.02771 (2019).
[2] Barni, Mauro, Kassem Kallas, and Benedetta Tondi. "A new backdoor attack in cnns by training set corruption without label poisoning." 2019 IEEE International Conference on Image Processing (ICIP). IEEE, 2019.
[3] Liu, Yunfei, et al. "Reflection backdoor: A natural backdoor attack on deep neural networks." Computer Vision–ECCV 2020: 16th European Conference, Glasgow, UK, August 23–28, 2020, Proceedings, Part X 16. Springer International Publishing, 2020.

**Questions:**

Could the authors explain more about how to choose \alpha and \beta? Are the two hyperparameters are attack-dependent or dataset-dependent or architecture-dependent? It is better to do some ablation studies.

Is the proposed method sensitive to different network architectures? It is better to show results on the same dataset using different architectures.

---

> ### Author Response · Authors · 2023-11-18
> **Response to Reviewer u6kT (Part 1/2)**
>
> **Q1. Experiments on More Baseline Attack Methods.**
>
> Thanks for the valuable suggestions. We provide the following experimental results on the mentioned clean-label backdoor attack methods (Refool[3], SIG[2], and Label-clean[1]). Each cell in the following table contains three values, which are precision (P), recall (R), and AUROC, respectively.
>
> | Attack $\downarrow$     | STRIP        | Frequency   | LAVA          |  Scale-UP          | CaBBD          |
> | ------------- |-------------| ---------| ------------- |---------| ------------- |
> | Refool[1] | 0.45,0.32,0.52 | 0.50,**1.00**,0.81 | 0.62,0.69,0.36 | 0.80, 0.87, 0.85 | **0.83**,0.88,**0.88** |
> | SIG[2] |  0.52,0.99,0.84 | 0.43,0.50,0.58 | 0.56,0.60,0.85 | 0.32,0.45,0.48 | **0.80**,**1.00**,**0.90**|
> | Label-clean[3] | **0.89**,0.45,0.91 | 0.5, 0.95, 0.85| 0.67,0.61,0.74| 0.76,0.96,0.91 | 0.82,**0.97**,**0.96** |
>
> As the table shows, our method generally has a significant advantage over the other baseline defense methods on the three metrics. We have added the experimental results in our updated manuscript. **It is also noted that a full table of added baseline attack methods is provided in Table 1 at the Global Response.**
>
> **Q2. Some obvious typos.**
>
> Thanks for your valuable comments. We have revised the typos in the updated manuscript.

---

> ### Author Response · Authors · 2023-11-18
> **Response to Reviewer u6kT (Part 2/2)**
>
> **Q3. More explanations on how to choose $\alpha$ and $\beta$.**
>
> Thanks for the insightful comment. In our setting, $\alpha$ and $\beta$ are not hyper-parameters that need to be tuned but some fixed parameters that can be pre-defined. Specifically, $\alpha$ value is fixed as 1 and $\beta$ value is fixed as $|S|-1$. We will answer the reasoning behind the choice with the following two aspects:
>
> Firstly, the choice is highly driven by our causality analysis in Section 2. Specifically, the sample-specific backdoor samples will show an abrupt prediction flipping if a small amount of noise is added, while sample-agnostic backdoor samples will show a consistent prediction until a considerably large noise is added. Therefore, $\alpha=1$ value is set to capture the sample-specific backdoor samples, and $\beta=|S|-1$ value is set to capture the sample-agnostic backdoor samples. We argue that setting $\alpha$ and $\beta$ to other values may lead to misalignment with the causality analysis.
>
> Secondly, the provided experiments across Section 4 all demonstrate that this choice is able to perform well on all three datasets against four types of backdoor attacks.
>
> To sum up, **we can readily use fixed value $\alpha = 1$ and $\beta= |S|-1$ when applying our backdoor detection method in practice.**
>
> Despite that $\alpha$ and $\beta$ are fixed values, we can tune the choice of magnitude set $|S|$ to achieve a better performance. Specifically, there are two hyper-parameters corresponding to the choice of magnitude set, i.e., step length and magnitude length. We conduct extensive experiments against all backdoor attack methods over three datasets to evaluate the performance of our method with different choices of step length and magnitude length. A quick view of the result is https://ibb.co/m0m97Lm. where the x-axis (horizontal) denotes the step_length, the y-axis (vertical) denotes the length of the magnitude set, and the value in each cell denotes the AUC score. It is seen that the method performs well when we choose the magnitude set with step_length=0.2 and length=7. These results have been added to the Appendix G of the updated manuscript.
>
> **Q4. Sensitive to Different Network Architectures**
>
> Really thanks for the insightful comment. We first argue that our method is model-agnostic and architecture-agnostic. As the setting suggests, our method only depends on two sources of information: (1) input image and (2) prediction labels from DNNs. Therefore, we do not assume any prior information about the model architectures. Besides, we have actually also evaluated our method with different architectures in our experiment. For example, we adopt ResNet architecture for Cifar10 and GTSRB datasets, and EfficientNet architecture for ImageNet-subset dataset. All the experiments have already validated the effectiveness of our method. That being said, we still value your suggestion as this is a very good way to further demonstrate the applicability of our method. We will add more experiments in the updated manuscript.
>
>
>
> [1] Turner, Alexander, Dimitris Tsipras, and Aleksander Madry. "Label-consistent backdoor attacks." arXiv preprint arXiv:1912.02771 (2019).
>
> [2] Barni, Mauro, Kassem Kallas, and Benedetta Tondi. "A new backdoor attack in cnns by training set corruption without label poisoning." 2019 IEEE International Conference on Image Processing (ICIP). IEEE, 2019.
>
> [3] Liu, Yunfei, et al. "Reflection backdoor: A natural backdoor attack on deep neural networks." Computer Vision–ECCV 2020: 16th European Conference, Glasgow, UK, August 23–28, 2020, Proceedings, Part X 16. Springer International Publishing, 2020.

---

> ### Author Response · Authors · 2023-11-20
> **Response to Reviewer u6kT**
>
> Dear Reviewer,
>
> Thanks for your assessment and your encouragement of our work. As the deadline for reviewer-author discussion (11/22/23) approaches, we would be happy to take this opportunity to make more discussions about any of our questions or concerns. If our response has addressed your concerns, we would be grateful if you could re-evaluate our paper based on our feedback and new supplementary results.
>
> Sincerely,
>
> Authors

---

> ### Author Response · Authors · 2023-11-21
> **Looking forward to your reply!**
>
> Dear Reviewer,
>
> Thanks for your assessment and your encouragement of our work. As the **deadline for reviewer-author discussion (11/22/23) approaches**, we would be happy to take this opportunity to make more discussions about any of our questions or concerns. If our response has addressed your concerns, we would be grateful if you could re-evaluate our paper based on our feedback and new supplementary results.
>
> Sincerely,
>
> Authors

---

> > ### Comment · Reviewer_u6kT · 2023-11-23
> >
> > Thank you for the rebuttal. Most of concerns have been addressed and I will keep the score.

---

### Official Review · Reviewer_PHNh · 2023-10-31

**Soundness:** 2 fair
**Presentation:** 3 good
**Contribution:** 2 fair
**Rating:** 6
**Confidence:** 3

**Summary:**

This study focuses on the problem of identifying backdoor samples. The authors introduced a framework for backdoor detection for cases where a separate clean validation dataset is unavailable. Their methodology draws from techniques found in the causal inference literature. The proposed methods are subject to experimental evaluation, using three distinct datasets.

**Strengths:**

1. Addressing the problem of backdoor sample detection is important and the introduction of Causality-based techniques for defense is new, at least to me.
2. The paper is well-written and easy to follow.
3. The experimental results seem to be promising.

**Weaknesses:**

1. The proposed method has undergone testing with only four types of attacks, which may not provide sufficient evidence to establish its effectiveness convincingly. It is recommended that the authors consider assessing its performance against a broader range of attacks, including adaptive backdoor attacks such as TaCT [a] and Adaptive Blend [b], as well as non-poisoning based backdoor attacks.
2. Is there a theoretical rationale for utilizing counterfactual samples, even when dealing with a simple linear model? While the empirical observations provide some insight, it is essential to gain a deeper analytical understanding of the underlying mechanisms that drive the proposed method.


Refs:
[a] Tang et al., "Demon in the Variant: Statistical Analysis of DNNs for Robust Backdoor Contamination Detection."
[b] Qi et al., "Revisiting the Assumption of Latent Separability for Backdoor Defenses."

**Questions:**

Please see the comments above.

---

> ### Author Response · Authors · 2023-11-18
> **Response to Reviewer PHNh**
>
> **Q1. Experiments with more baseline attack methods**
>
> Thanks for the valuable suggestions. We provide the following experimental results on the mentioned backdoor attack methods (TaCT[1] and Adaptive Blend[2]). Each cell in the following table contains three values, which are precision (P), recall (R), and AUROC, respectively.
>
> | Attack $\downarrow$     | STRIP        | Frequency   | LAVA          |  Scale-UP          | CaBBD          |
> | ------------- |-------------| ---------| ------------- |---------| ------------- |
> | TaCT[1] | 0.47,0.96,0.92| 0.50,**0.98**,0.50 | 0.40, 0.42, 0.60 | 0.61,0.70,0.62 | **0.85**,0.80,**0.97**|
> | Adaptive Blend[2] | 0.50,**1.00**,0.94| 0.50,**1.00**,0.50 | 0.45, 0.46, 0.64 | 0.58,0.74,0.62 | **0.81**,0.73,**0.96**|
>
> As the table shows, our method generally has a significant advantage over the other baseline defense methods on the three metrics. We have added the experimental results in our updated manuscript. **It is also noted that a full table of added baseline attack methods is provided in Table 1 at the Global Response.**
>
> **Q2. Theoretical Rationale for Utilizing Counterfactual Samples.**
>
> Thanks for the insightful suggestion. Firstly, we argue that introducing counterfactual samples is not a randomly proposed strategy, but is highly supported by the theoretical causal analysis in Section 2 and highly motivated by a real-life example of identifying salt and sugar provided in Section 2.3. That being said, we agree that adding more mathematical justifications on why we utilize counterfactual samples could make the paper more theoretically sound. Therefore, we will work on this and provide our theoretical results in the updated manuscript.
>
> [1] Tang et al., "Demon in the Variant: Statistical Analysis of DNNs for Robust Backdoor Contamination Detection."
>
> [2] Qi et al., "Revisiting the Assumption of Latent Separability for Backdoor Defenses."

---

> ### Author Response · Authors · 2023-11-20
> **Response to Reviewer PHNh**
>
> Dear Reviewer,
>
> Thanks for your assessment and your encouragement of our work. As the deadline for reviewer-author discussion (11/22/23) approaches, we would be happy to take this opportunity to make more discussions about any of our questions or concerns. If our response has addressed your concerns, we would be grateful if you could re-evaluate our paper based on our feedback and new supplementary results.
>
> Sincerely,
>
> Authors

---

> ### Author Response · Authors · 2023-11-21
> **Looking forward to your reply!**
>
> Dear Reviewer,
>
> Thanks for your assessment and your encouragement of our work. As the **deadline for reviewer-author discussion (11/22/23) approaches**, we would be happy to take this opportunity to make more discussions about any of our questions or concerns. If our response has addressed your concerns, we would be grateful if you could re-evaluate our paper based on our feedback and new supplementary results.
>
> Sincerely,
>
> Authors

---

> > ### Comment · Reviewer_PHNh · 2023-11-22
> > **Thanks for the authors' rebuttal**
> >
> > I would like to thank the authors for doing extra experiments and addressing my concerns about the theory. They've taken care of the issues with the experimental results, but the theoretical ones still need some attention. Based on the effectiveness of this method, I will increase my score.

---

> > > ### Author Response · Authors · 2023-11-22
> > > **Sincere Thanks to Reviewer PHNh**
> > >
> > > Thank you so much for improving the scores. It encourages us a lot!
> > >
> > > Sincerely,
> > >
> > > Authors

---

### Official Review · Reviewer_k4uD · 2023-11-03

**Soundness:** 3 good
**Presentation:** 2 fair
**Contribution:** 2 fair
**Rating:** 3
**Confidence:** 4

**Summary:**

This paper proposes a method to detect backdoor samples at
run-time in the black-box scenario. The proposed method is based
on the causality analysis of the backdoor attacks. It works
by adding the noises under different magnitudes on the
examined images and detects backdoor samples by analyzing
the prediction's sensitivity to the magnitudes of the added
noise. Experiments demonstrate that the proposed method is
effective.

**Strengths:**

* Run-time backdoor sample detection in the black-box manner is an important problem.

* Both sample-agnostic attacks and sample-specific attacks are analyzed and discussed.

**Weaknesses:**

* The novelty of this paper might be limited as this paper
claims to be the first to analyze backdoor predictions from
a causal perspective, while previous work by Zhang et al.
[1] has conducted a similar analysis. This paper lacks a
detailed comparison between Zhang et al.'s causal analysis
and its own although it cites Zhang et al. The connection
between the proposed method and the causal analysis is not
clear. The definition of counterfactual samples and the
rationale behind considering noise-added samples as
counterfactual is vague. The proposed method distinguishes
backdoor samples and clean samples based on their different
sensitivity to the perturbations or augmentations, which
shares similar spirits to many existing methods such as
STRIP.

* There are some attacks that are robust to the
perturbations, such as Wang et al. [2]. The
color-style-based attacks [3,4,5] might also have stronger
robustness to the added noises compared to the attacks
involved in the experiments. The evaluation of these attacks
is missing. In addition, this paper only uses one simple
trigger pattern for the BadNet attack. It is suggested to
use more complicated patterns with large pixel values to
make the evaluation more comprehensive.

* This paper does not explore adaptive attacks where
attackers are aware of the defense mechanism and
actively attempt to circumvent it.


[1] Zhang et al., Backdoor Defense via Deconfounded Representation Learning. CVPR 2023.

[2] Wang et al., Robust Backdoor Attack with Visible, Semantic, Sample-Specific, and Compatible Triggers. arXiv 2023.

[3] Jiang et al., Color Backdoor: A Robust Poisoning Attack in Color Space. CVPR 2023.

[4] Cheng et al., Deep Feature Space Trojan Attack of Neural Networks by Controlled Detoxification. AAAI 2021.

[5] Liu et al., ABS: Scanning Neural Networks for Back-doors by Artificial Brain Stimulation. CCS 2019.

**Questions:**

See Weaknesses.

---

> ### Author Response · Authors · 2023-11-18
> **Response to Reviewer k4uD (Part 1/3)**
>
> **Q1. The comparison with Zhang et al. [1]**
>
> Thanks for the insightful question. We argue that there are several fundamental differences between the causal analysis in our paper and that in [1].
>
> - [**Analysis Objective**]: Their analysis aims to provide a theoretical analysis for training a clean model from a backdoored dataset, while our analysis aims to investigate the distinct prediction behaviors of clean and backdoored images from a causal perspective.
>
> - [**Analysis Content**]: Their analysis uses causal graphs to model the generation process of backdoor data in the **training stage**, while our analysis focuses on DNNs' prediction behaviors in the **inference stage** when input with different types of data. Although similar structures are used (e.g., $I \leftarrow A \rightarrow Y$), the actual meaning of each edge is fundamentally different. For example, in [1], $A \rightarrow Y$ means backdoor attackers will "change the labels to the targeted label" when constructing backdoor samples (evidenced by Section 3.2 of [1]), but in our setting, $A \rightarrow Y$ means that backdoor attacks will make the backdoored DNNs predict the backdoor image as label $Y$.
>
> - [**Analysis Usage**]: Their analysis is used to adjust confounder, by disentangling backdoor path and causal path in the model training process. However, our causal analysis works as a guideline for distinguishing backdoor samples and clean samples in the inference stage.
>
> **To sum up, there are fundamental differences between the two papers.** We have also added a detailed comparison of the two papers in the updated manuscript in Appendix A.
>
> **Q2. The connection between the proposed method and the causal analysis.**
>
> We argue that the proposed method in Section 3 is **highly supported** by the causal analysis in Section 2. The connection between the two parts is summarized as follows:
> - [**Intuition: Causality analysis**] Causality analysis provides an intuition for distinguishing backdoor and clean samples. As shown in the causal graphs (Figure 2), predictions on backdoor samples and clean samples are determined by different paths in the causal graphs. Therefore, an intuitive strategy for distinguishing the two types of samples is to assess whether the DNN's prediction is
> primarily influenced by the direct causal path or the spurious path.
>
> - [**Challenge: Causality is invisible**] However, the black-box nature of DNNs and the limited available information in our setting render this direct causal analysis challenging.
>
> - [**Strategy: Introducing counterfactual examples as an intervention**] Therefore, we propose an indirect method that implicitly induces some distinguishable behaviors of backdoor samples and clean samples. Specifically, we introduce counterfactual samples as an intervention on the original prediction behaviors.
>
> It is also noted that this strategy is not a randomly proposed one, but **highly motivated** by a real-life example of identifying salt and sugar in Figure 3: *Distinct molecular structures (**causality**) determine whether a substance is sugar or salt. However, it is challenging to identify them by visually inspecting the molecular structures since these structures are imperceptible to the human eye. Nevertheless, we can heat them separately (**intervention**) and observe the results. Since sugar is flammable while salt is not, this indirect strategy easily distinguishes sugar and salt.*
>
> **Q3. The definition of counterfactual samples.**
>
> Sorry for the confusion. The definition of counterfactual samples in our paper follows the prior well-established works [4,5,6], which generally means the manipulation of the real (factual) image.

---

> ### Author Response · Authors · 2023-11-18
> **Response to Reviewer k4uD (Part 2/3)**
>
> **Q4. The rationale behind considering noise-added samples as counterfactual.**
>
> Thanks for the question. There are many popular counterfactual generation methods in the vision domain, including noise-based [5,6], random masking [4], and mix-up [7]. We have evaluated the performance with different counterfactual generation method and found that **noise-based outperform the others in our problem setting**. The corresponding results are reported in Table 2.
>
> **Q5. Comparison with STRIP.**
> Thanks for the insightful question. It is admitted that our method shares some extent of similar spirit with STRIP. However, the differences between the two methods are also significant:
>
> - [**Assumptions**] STRIP assumes access to the probability logits of DNNs and additional validation datasets. However, our method assumes only black-box access to the model's prediction labels and none of the validation set. Therefore, our problem aims at solving a problem that is intrinsically more challenging and more practical in the real world.
>
> - [**Performance**] As reported in Table 1, Our method performs better than STRIP in most of the cases despite its additional assumptions on probability logits and validation set.
>
> **Q6. Experiments with more baseline attack methods.**
>
> Thanks for the valuable suggestions. We provide additional results on two open-sourced attack methods [2] and [3] on Cifar10 dataset. The following table compares the performance of our method (CaBBD) with other baseline defense methods, where each cell contains three numbers, denoting Precision (P), Recall (R), and AUROC, respectively.
>
>
> | Attack $\downarrow$     | STRIP      | LAVA   | Frequency          |  Scale-UP          | CaBBD (Ours)         |
> | ------------- |-------------| ---------| ------------- | ---------| ------------- |
> | DFST[2] | 0.51, 0.99, 0.70 | 0.67,0.54,0.72 | 0.5,1.0,0.45 | 0.46,0.55,0.57 | **0.78**,**0.99**,**0.90**|
> | Filter[3] | 0.48,0.35,0.54 | 0.44,0.68,0.70 | 0.5,**0.99**,0.50 | 0.49,0.65,0.51 | **0.64**,**0.99**,**0.80**|
>
> As shown, our method has a significant advantage over the baseline defense methods in all three metrics. We have added the experimental results in our updated manuscript. **It is also noted that a full table of added baseline attack methods is provided in Table 1 at the Global Response.**

---

> ### Author Response · Authors · 2023-11-18
> **Response to Reviewer k4uD (Part 3/3)**
>
> **Q7. Experiments with more complicated BadNet patterns.**
>
> Thanks for the suggestions. In the default setting, we use grid-like $3\times3$ pixel squares as the trigger pattern(Figure 9). Moreover, we have also compared the performance of our method when the size of the BadNet trigger patterns varies (e.g., $2\times 2, 4\times 4, 5\times 5, ...)$ in Figure 7. To further demonstrate the effectiveness of our method against more complicated BadNet trigger patterns, we provide some additional results on BadNet when the trigger pattern is randomly generated by `torch.rand(3,3)`.
>
> | Attack $\downarrow$     | STRIP      | LAVA   | Frequency          |  Scale-UP          | CaBBD (Ours)         |
> | ------------- |-------------| ---------| ------------- | ---------| ------------- |
> | BadNet-randomly-generated | 0.83, 0.99, 0.95 | 0.73,0.66,0.79 | 0.71,0.78,0.81 | **0.88**,**0.98**,**0.92** | **0.88**,**0.98**,0.90|
>
>
> **Q8. Adaptive attacks analysis.**
>
> Thanks for the valuable suggestion. Suppose an attacker already knows our defense methods in advance, then intuitively the attacker will train the DNN model with the following adaptive training loss function:
>
> $\min_{\theta} \sum_{i=1}^{|D_{/c}|} \ell (f_{\theta}(x_i ), y_i) + \sum_{i=1}^{|D_b|} \ell (f_{\theta}(x_i'), y_t) +  \sum_{i=1}^{|D_b|} \ell (f_{\theta}(x_i' + m * \epsilon_i), y_i)$
>
> where $\epsilon_i \sim N(0,1)$ is a random Gaussian noise added to each backdoor sample $x_i'$, and $m$ is the magnitude multiplier of the added noise. Intuitively, this loss function will make the prediction behaviors on backdoored images (when the noise is added) more similar to those on clean images. To evaluate whether our method is still effective against adaptive attacks, we provide results with varying $m$ against BadNet attack on the Cifar10 dataset in the following table.
>
> | Noise Mangitude Multiplier $m$  | 0 | 0.1 | 0.5   | 1.0          |  1.5         |
> | ------------- |-------------| ---------| ------------- | ---------| ---------|
> | (Precision, Racall, AUC) $\rightarrow$ | 0.85, 1.00 0.91 | 0.77,0.91,0.85 | 0.72,0.94,0.87 | 0.30,0.42,0.64| 0.24,0.34,0.55|
>
> It can be inferred that our method can still achieve a satisfactory performance when the noise magnitude is small (e.g. < 0.5), but the performance will gradually drop when the noise magnitude multiplier becomes higher (e.g., > 1.0). However, we argue that when the noise magnitude multiplier > 1.0, the whole image will be dominated by random noise, rendering the modified image easily detected and filtered out by the off-the-shelf backdoor filtering algorithms.
>
> [1] Zhang, Z., Liu, Q., Wang, Z., Lu, Z., & Hu, Q. (2023). Backdoor Defense via Deconfounded Representation Learning. In Proceedings of the IEEE/CVF Conference on Computer Vision and Pattern Recognition (pp. 12228-12238).
>
> [2] Cheng, S., Liu, Y., Ma, S., & Zhang, X. (2021, May). Deep feature space trojan attack of neural networks by controlled detoxification. In Proceedings of the AAAI Conference on Artificial Intelligence (Vol. 35, No. 2, pp. 1148-1156).
>
> [3] Liu, Y., Lee, W. C., Tao, G., Ma, S., Aafer, Y., & Zhang, X. (2019, November). Abs: Scanning neural networks for back-doors by artificial brain stimulation. In Proceedings of the 2019 ACM SIGSAC Conference on Computer and Communications Security (pp. 1265-1282).
>
> [4] Xiao, Y., Tang, Z., Wei, P., Liu, C., & Lin, L. (2023). Masked Images Are Counterfactual Samples for Robust Fine-tuning. In Proceedings of the IEEE/CVF Conference on Computer Vision and Pattern Recognition (pp. 20301-20310).
>
> [5] Jeanneret, G., Simon, L., & Jurie, F. (2023). Adversarial Counterfactual Visual Explanations. In Proceedings of the IEEE/CVF Conference on Computer Vision and Pattern Recognition (pp. 16425-16435).
>
> [6] Chang, C. H., Creager, E., Goldenberg, A., & Duvenaud, D. (2018). Explaining image classifiers by counterfactual generation. arXiv preprint arXiv:1807.08024.
>
> [7] Yu, L., Mao, Y., Wu, J., & Zhou, F. (2023, July). Mixup-based unified framework to overcome gender bias resurgence. In Proceedings of the 46th International ACM SIGIR Conference on Research and Development in Information Retrieval (pp. 1755-1759).

---

> ### Author Response · Authors · 2023-11-20
> **Response to Reviewer k4uD**
>
> Dear Reviewer,
>
> Thanks for your assessment and your encouragement of our work. As the deadline for reviewer-author discussion (11/22/23) approaches, we would be happy to take this opportunity to make more discussions about any of our questions or concerns. If our response has addressed your concerns, we would be grateful if you could re-evaluate our paper based on our feedback and new supplementary results.
>
> Sincerely,
>
> Authors

---

> ### Author Response · Authors · 2023-11-21
> **Looking forward to your reply!**
>
> Dear Reviewer,
>
> Thanks for your assessment and your encouragement of our work. As the **deadline for reviewer-author discussion (11/22/23) approaches**, we would be happy to take this opportunity to make more discussions about any of our questions or concerns. If our response has addressed your concerns, we would be grateful if you could re-evaluate our paper based on our feedback and new supplementary results.
>
> Sincerely,
>
> Authors

---

> ### Comment · Reviewer_k4uD · 2023-11-23
>
> Thanks for your detailed responses. I still have the following concerns after reading the responses:
>
> * The concerns about the novelty of this paper have not been addressed. The high-level idea of the proposed method (backdoor samples and clean samples have different sensitivity to the perturbations or augmentations) shares similar spirits to many existing methods such as STRIP. I know the detailed implemetation of this method is different to that of STRIP, but the high-level idea might be similar.
>
> * The proposed method might be not general to different types of attacks. The results show that the proposed method can not effectively detect the backdoor samples of the color-based attacks. For example, the detection precision on filter attack is only 64%. There are many existing methods that can effectively detect the backdoor samples in filter attack, such as MISA (Kiourti et al., 2021)
> and Beatrix (Ma et al., 2023). The evaluation on the semantic-based attack such as Wang et al. and Lin et al. is still missing.
>
> * For the adaptive attack, since the strong random Gaussian noise is added in the training stage and the attacker can control the training of the backdoored model (in the threat model of the run-time backdoor sample detection problem). The authors mention that "modified image easily detected and filtered out by the off-the-shelf backdoor filtering algorithms". Why would the model trainer, who is also the attacker, choose to filter out these altered images?
>
> Based on these remaining concerns, I am inclined to maintain my current score and provide the space for further improvement of this work.
>
> Kiourti et al., MISA: Online Defense of Trojaned Models using Misattributions. in ACSAC 2021
>
> Ma et al., The “Beatrix” Resurrections: Robust Backdoor Detection via Gram Matrices. in NDSS 2023
>
> Wang et al., Robust Backdoor Attack with Visible, Semantic, Sample-Specific, and Compatible Triggers. arXiv 2023
>
> Lin et al., Composite Backdoor Attack for Deep Neural Network by Mixing Existing Benign Features. in CCS 2020

---

### Author Response · Authors · 2023-11-18
**To All Reviewers**

Dear Reviewers,

Thanks very much for your constructive suggestions and for appreciating the novelty of our work. We respectfully accept your valuable suggestions and added more experiments.

To address your major concerns about the experiments, we try our best to conduct additional experiments and update the manuscript, including
- A more detailed comparison between our method and Zhang et al. [1]. (**Updated to Appendix A**)
- Additional baseline attacks (DFST, Filter, TaCT, Adaptive Blend, Refool, SIG, and Label-clean), showing that CaBBD can be effective against a wide range of backdoor attack methods (**Updated to Appendix K**)

**Table 1. Evaluation against More Attack Baseline Methods on Cifar10.**

| Attack $\downarrow$     | STRIP      | LAVA   | Frequency          |  Scale-UP          | CaBBD (Ours)         |
| ------------- |-------------| ---------| ------------- | ---------| ------------- |
| DFST | 0.51, 0.99, 0.70 | 0.67,0.54,0.72 | 0.5,1.0,0.45 | 0.46,0.55,0.57 | **0.78**,**0.99**,**0.90**|
| Filter | 0.48,0.35,0.54 | 0.44,0.68,0.70 | 0.5,**0.99**,0.50 | 0.49,0.65,0.51 | **0.64**,**0.99**,**0.80**|
| TaCT | 0.47,0.96,0.92| 0.50,**0.98**,0.50 | 0.40, 0.42, 0.60 | 0.61,0.70,0.62 | **0.85**,0.80,**0.97**|
| Adaptive Blend | 0.50,**1.00**,0.94| 0.50,**1.00**,0.50 | 0.45, 0.46, 0.64 | 0.58,0.74,0.62 | **0.81**,0.73,**0.96**|
| Refool | 0.45,0.32,0.52 | 0.50,**1.00**,0.81 | 0.62,0.69,0.36 | 0.80, 0.87, 0.85 | **0.83**,0.88,**0.88** |
| SIG |  0.52,0.99,0.84 | 0.43,0.50,0.58 | 0.56,0.60,0.85 | 0.32,0.45,0.48 | **0.80**,**1.00**,**0.90**|
| Label-clean | **0.89**,0.45,0.91 | 0.5, 0.95, 0.85| 0.67,0.61,0.74| 0.76,0.96,0.91 | 0.82,**0.97**,**0.96** |

- Adding more results on more complicated BadNet trigger patterns.


| Attack $\downarrow$     | STRIP      | LAVA   | Frequency          |  Scale-UP          | CaBBD (Ours)         |
| ------------- |-------------| ---------| ------------- | ---------| ------------- |
| BadNet-randomly-generated | 0.83, 0.99, 0.95 | 0.73,0.66,0.79 | 0.71,0.78,0.81 | **0.88**,**0.98**,**0.92** | **0.88**,**0.98**,0.90|


- Adding discussions about adaptive attacks, and discussing what strategies would be used when the attackers have prior information about the CaBBD defense method. (**Updated to Appendix J**)


- Adding more experiments about the choice of magnitude set $|S|$. (**Updated to Appendix G**).


[1] Zhang, Z., Liu, Q., Wang, Z., Lu, Z., & Hu, Q. (2023). Backdoor Defense via Deconfounded Representation Learning. In Proceedings of the IEEE/CVF Conference on Computer Vision and Pattern Recognition (pp. 12228-12238).

---

### Author Response · Authors · 2023-11-23
**A Gentle Reminder of the Final Feedback**

Dear Reviewers,

Thanks for your assessment and your encouragement of our work. As the deadline for reviewer-author discussion (11/22/23) approaches, we would be happy to take this opportunity to make more discussions about any of our questions or concerns. If our response has addressed your concerns, we would be grateful if you could re-evaluate our paper based on our feedback and new supplementary results.

Sincerely,

Authors

---

### Meta-Review · Area_Chair_zUo2 · 2023-12-10

**Metareview:**

This work studied the black-box backdoor detection task inspired by causality analysis.

Most reviewers recognized the importance of the studied task, and the good writing.
Meanwhile, several important concerns were also proposed, including: 1. The novelty of the causality analysis and the behind analysis. The authors provided some explanations about the differences between this work and Zhang et al. [1]. More comprehensive discussions, not only difference but also connections should be discussed. More important, the authors claimed sec 2.3 provided theoretical analysis, but the fact is that there is only description about figure 2, without solid justification.
2. The connection between the causality perspective and the proposed method is unclear. The authors didn’t give a direct and satisfied answer during the rebuttal.
3. The experiments are very insufficient, considering the evaluated attack methods, architectures, etc. The authors added a few suggested methods, but refused to add some important suggested methods. The effectiveness of the proposed method is questionable.

Hope all reviews are helpful to further improve this work.

**Justification For Why Not Higher Score:**

see above

**Justification For Why Not Lower Score:**

n/a

---

### Decision · Program_Chairs · 2024-01-16

Reject